# Multi-day neuron tracking in high-density electrophysiology recordings using earth mover's distance

**Augustine Xiaoran Yuan[1,2], Jennifer Colonell[1], Anna Lebedeva[3], Michael Okun[4], Adam S Charles[2]\*, Timothy D Harris[1,2]\***

[1]Janelia Research Campus, Howard Hughes Medical Institute, Ashburn, United States; [2]Department of Biomedical Engineering, Center for Imaging Science Institute, Kavli Neuroscience Discovery Institute, Johns Hopkins University, Baltimore, United States; [3]Sainsbury Wellcome Centre, University College London, London, United Kingdom; [4]Department of Psychology and Neuroscience Institute, University of Sheffield, Sheffield, United Kingdom

**\*For correspondence:**
adamsc@jhu.edu (ASC);
harrist@janelia.hhmi.org (TDH)

**Competing interest:** The authors declare that no competing interests exist.

**Abstract** Accurate tracking of the same neurons across multiple days is crucial for studying changes in neuronal activity during learning and adaptation. Advances in high-density extracellular electrophysiology recording probes, such as Neuropixels, provide a promising avenue to accomplish this goal. Identifying the same neurons in multiple recordings is, however, complicated by non-rigid movement of the tissue relative to the recording sites (drift) and loss of signal from some neurons. Here, we propose a neuron tracking method that can identify the same cells independent of firing statistics, that are used by most existing methods. Our method is based on between-day non-rigid alignment of spike-sorted clusters. We verified the same cell identity in mice using measured visual receptive fields. This method succeeds on datasets separated from 1 to 47 days, with an 84% average recovery rate.

## eLife assessment

This **important** study proposes a new method for tracking neurons recorded with Neuropixel electrodes across days. The methods and the strength of the evidence are **convincing**, but the authors do not address whether their approach can be generalized to other brain areas, species, behaviors, or tools. Overall, this method will be potentially of interest to many neuroscientists who want to study long-term activity changes of individual neurons in the brain.

## Introduction

The ability to longitudinally track neural activity is crucial to understanding central capabilities and changes of neural circuits that operate on long time-scales, such as learning and plasticity (*Carmena et al., 2005*; *Clopath et al., 2017*; *Huber et al., 2012*; *Liberti et al., 2016*), motor stability (*Carmena et al., 2005*; *Dhawale et al., 2017*; *Jensen et al., 2022*), etc. We seek to develop a method capable of tracking single units regardless of changes in functional responses for the duration of an experiment spanning 1–2 months.

High-density multi-channel extracellular electrophysiology (ephys) recording devices enable chronic recordings over large areas for days to months (*Steinmetz et al., 2021*). Such chronic recordings make possible experiments targeted at improving our understanding of neural computation and underlying mechanisms. Examples include perceptual decision-making, exploration, and navigation

**Figure 1.** Schematic depiction of drift. (**a**) Mice were implanted with a four-shank Neuropixels 2.0 probe in visual cortex area V1. (**b**) Each colored star represents the location of a unit recorded on the probe. In this hypothetical case, the same color indicates unit correspondence across days. The black unit is missing on day 48, while the turquoise star is an example of a new unit. Tracking aims to correctly match the red and blue units across all datasets and determine that the black unit is undetected on day 48. (**c**) Two example spatial-temporal waveforms of units recorded in two datasets that likely represent the same neuron, based on similar visual responses. Each trace is the average waveform on one channel across 2.7 ms. The blue traces are waveforms on the peak channel and nine nearby channels (two rows above, two rows below, and one in the same row) from the first dataset (day 1). The red traces, similarly selected, are from the second dataset. Waveforms are aligned at the electrodes with peak amplitude, different on the 2 days.

(*Brown et al., 2004*; *Buzsáki, 2004*; *Harris, 2005*; *Harris et al., 2016*; *Luo et al., 2020*; *Quian Quiroga and Panzeri, 2009*). Electrode arrays with hundreds to thousands of sites, e.g., Neuropixels, are now used extensively to record the neural activity of large populations stably and with high spatio-temporal resolution, capturing hundreds of neurons with single neuron resolution (*Buzsáki, 2004*; *Harris et al., 2016*). Moreover, ephys retains the higher time resolution needed for single spike identification, as compared with calcium imaging that provides more spatial cues with which to track neurons over days.

The first step in analyzing ephys data is to extract single neuron signals from the recorded voltage traces, i.e., spike sorting. Spike sorting identifies individual neurons by grouping detected action potentials using waveform profiles and amplitudes. Specific algorithms include principal components-based methods (*Quiroga et al., 2004*; *Chah et al., 2011*), and template-matching methods, e.g., Kilosort (*Brown et al., 2004*; *Carlson and Carin, 2019*; *Harris et al., 2016*; *Pachitariu et al., 2016*). Due to the high-dimensional nature of the data, spike sorting is often computationally intensive on large datasets (tens to hundreds of GB) and optimized to run on single sessions. Thus processing multiple sessions has received minimal attention, and the challenges therein remain largely unaddressed.

One major challenge in reliably tracking neurons is the potential for changes in the neuron population recorded (*Figure 1a* and *Figure 1b*). In particular, since the probe is attached to the skull, brain tissue can move relative to the probe, e.g., during licking, and drift can accumulate over time (*Jun et al., 2017*). Kilosort 2.5 corrects drift within a single recording by inferring tissue motion from continuous changes in spiking activity and interpolating the data to account for that motion (*Steinmetz et al., 2021*). Larger between-recording drift occurs for sessions on different days, and can (1) change the size and location of spike waveforms along the probe (*Hall et al., 2021*), (2) lose neurons

that move out of range, and (3) gain new neurons that move into recording range. Thus clusters can change firing pattern characteristics or completely appear/disappear. As a result the specific firing patterns classified as unit clusters may appear and disappear in different recordings (*Bar-Hillel et al., 2006*; *Harris et al., 2016*; *Swindale and Spacek, 2014*; *Tolias et al., 2007*). Another challenge is that popular template-matching-based spike-sorting methods usually involve some randomness in template initialization (*Chung et al., 2017*; *Lee et al., 2020*; *Pachitariu et al., 2016*). As a result, action potentials can be assigned into clusters differently, and clusters can be merged or separated differently across runs.

Previous neuron tracking methods are frequently based on waveform and firing statistics, e.g., firing rate similarity (*Chung et al., 2019*), action potential shape correlation, and inter-spike interval (ISI) histogram shape (*Vasil'eva et al., 2016*). When neuronal representations change, e.g., during learning (*Carmena et al., 2005*; *Huber et al., 2012*; *Liberti et al., 2016*) or representational drift (*Rokni et al., 2007*), neural activity statistics became less reliable. In this work, we take advantage of the rich spatial-temporal information in the multi-channel recordings, matching units based on the estimated neuron locations, and unit waveforms (*Lewicki, 1998*), instead of firing patterns.

As an alternative method, *Steinmetz et al., 2021*, concatenated pairs of datasets after low-resolution alignment, awkward for more than two datasets. We report here a more flexible, expand-able, and robust tracking method that can track neurons effectively and efficiently across any number of sessions.

## Results

### Procedure

Our datasets consist of multiple recordings taken from three mice (Figure 7a) over 2 months. The time gap between two recordings ranges from 2 to 25 days. Each dataset is spike-sorted individually with a standard Kilosort 2.5 pipeline. The sorting results, including unit assignment, spike times, etc., are used as input for our method (post-processed using ecephys spike sorting pipeline; *Colonell, 2018*) (section Dataset). To ensure the sorting results are unbiased, we performed no manual curation. As the clusters returned by Kilosort can vary in quality, we only considered the subset of units labeled as 'good' by Kilosort, here referred to as KSgood units (section Reference set). KSgood units are mainly determined by the amount of ISI violations and are believed to represent a single unit (*Pachitariu et al., 2016*).

Our overall strategy is to run spike sorting once per session, and then to generate a unit-by-unit assignment between pairs of datasets. When tracking units across more than two sessions, two strategies are possible: match all ensuing sessions to a single session (e.g. the first session) (section Measuring rigid drift using the EMD and section Determining *z*-distance threshold), or match consecutive pairs of sessions and then trace matched units through all sessions (section Units can be tracked in discontinuous recordings for 48 days).

We refer to the subset of KSgood units with strong and distinguishable visual responses in both datasets of a comparison as reference units (see section Reference set for details). Similar to *Steinmetz et al., 2021*, we validated our unit matching of reference units using visual receptive field similarity. Finally, we showed that trackable units with strong visual responses are qualitatively similar to those without (*Figure 5—figure supplement 1* to Figure 5).

To provide registration between pairs of recordings, we used the earth mover's distance (EMD) (*Bertrand et al., 2020*; *Cohen, 1999*). We use a feature space consisting of a geometric distance space and a waveform similarity space, to address both rigid and non-rigid neuron motion. The EMD finds matches between objects in the two distributions by minimizing the overall distances between the established matches (section Earth mover's distance).

We use EMD in two stages: rigid drift correction and unit assignment. Importantly, the EMD incorporates two parameters crucial for matching units: location-based physical distance and a waveform distance metric that characterizes similarity of waveforms (section Calculating the EMD metric). The EMD matrix is constructed with a weighted combination of the two (details in section Methods), i.e., a distance between two units $d_{ik}$ is given by $d_{ik} = d_{location_{ik}} + \omega * d_{waveform_{ik}}$ (*Figure 2a*). The first EMD stage estimates the homogeneous vertical movement of the entire population of KSgood units (*Figure 2b*). This movement estimate is used to correct the between-session rigid drift in unit

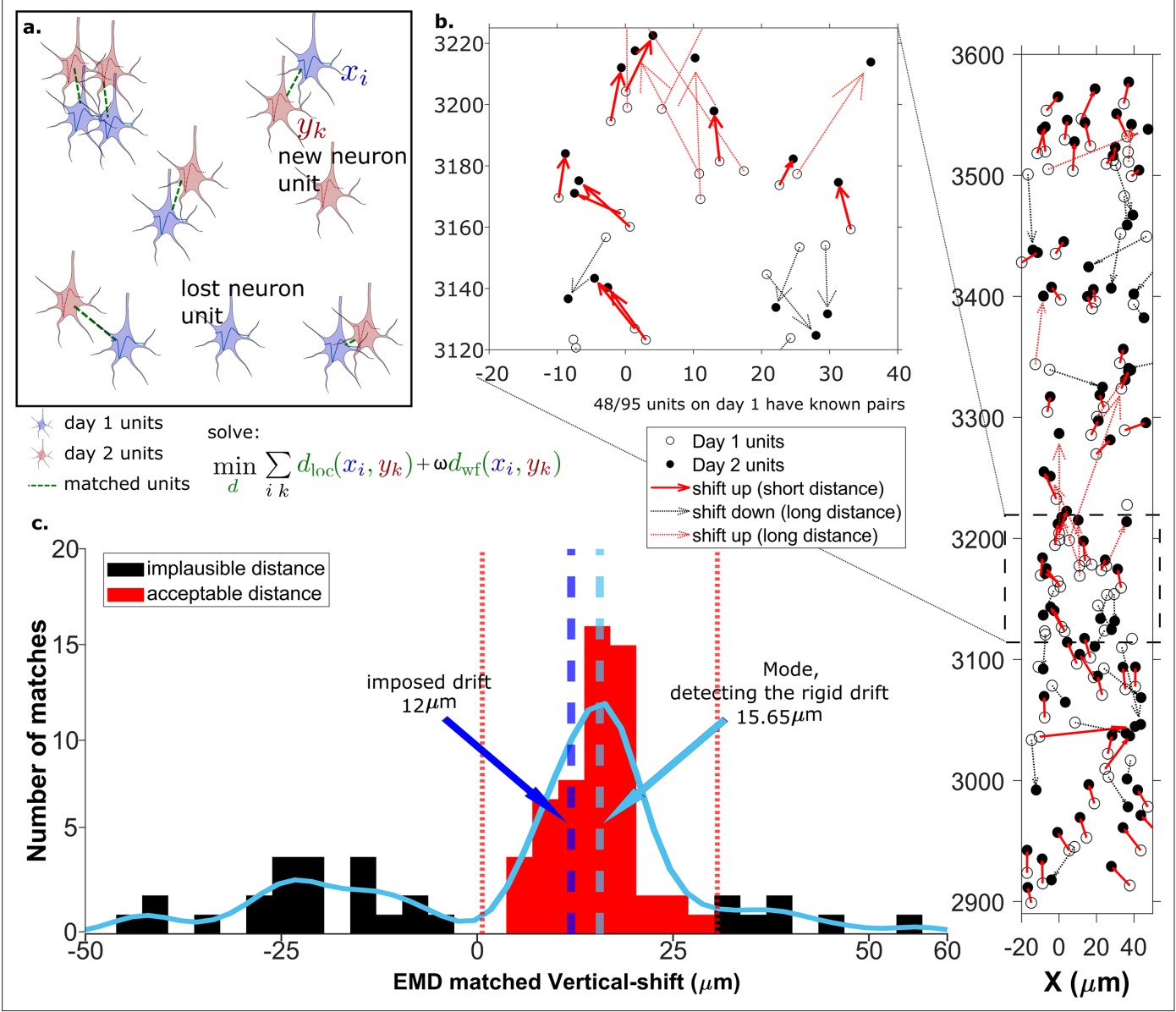

**Figure 2.** The earth mover's distance (EMD) can detect the displacement of single units. (**a**) Schematic of EMD unit matching. Each blue unit in day 1 is matched to a red unit in day 2. Dashed lines indicate the matches to be found by minimizing the weighted sum of physical and waveform distances. (**b**) Open and filled circles show positions of units in days 1 and 2, respectively. Arrows indicate matching using EMD. The arrow color represents the match direction; upward matches found with the EMD are in red and downward in black. Solid lines indicate a $z$-match distance within 15 µm, while a dashed line indicates a $z$-distance > 15µm. Expanded view shows probe area from 3120 to 3220 µm. (**c**) Histogram of $z$-distances of matches (black and red bars) and kernel fit (light blue solid curve). The light blue dashed line shows the mode ($d_m = 15.65$µm). The dark blue dashed line shows the imposed drift ($d_i = 12$µm). The red region shows the matches within 15 µm of the mode. The EMD needs to detect the homogeneous movement against the background, i.e., units in the black region that are unlikely to be the real matches due to biological constraints.

The online version of this article includes the following figure supplement(s) for figure 2:

**Figure supplement 1.** The effect of drift correction on reference units yield for all three animals.

**Figure supplement 2.** Earth mover's distance (EMD) cost can be used to detect discontinuities in the data.

**Figure supplement 3.** The normalized earth mover's distance (EMD) cost (unitless), $z$-distance (µm), physical distance (µm), and waveform distance (unitless) and the corresponding recovery rate of reference unit (units with matched visual responses) in pairwise matches of all to all pairs of recordings, on each shank.

**Figure supplement 4.** Recovery rate vs. L2-weight.

locations. The rigid drift estimation procedure is illustrated in *Figure 2b*. Post drift correction, a unit's true match will be close in both physical distance and waveform distance. Drift-corrected units were then matched at the second EMD stage. The EMD between assigned units can be thought of as the local non-rigid drift combined with the waveform distortion resulting from drift. We test the accuracy of the matching by comparing with reference unit assignments based on visual receptive fields (section Reference set).

For each unit, the location is determined by fitting the peak-to-peak amplitudes on the 10 sites nearest the site with peak signal, based on the triangulation method in *Boussard et al., 2021* (section Calculating the EMD metric). The waveform distance is an L2 norm between two spatial-temporal waveforms that spans 22 channels and 2.7 ms (section Calculating the EMD metric). Physical unit distances provide a way to maintain the internal structure and relations between units in the EMD. Waveform similarity metrics will distinguish units in the local neighborhood and likely reduce the effect of new and missing units.

We analyzed the match assignment results in two ways. First, we compared all subsequent datasets to dataset 1 using recovery rate and accuracy. We define recovery rate $R_{rec}$ as the fraction of unit assignments by our method that are the same as reference unit assignments established using visual responses (section Reference set).

$$P\left(EMD \mid ref\right) = \frac{P\left(EMD \cap ref\right)}{P\left(ref\right)} = \frac{N_{EMD \cap ref}}{N_{ref}} \tag{1}$$

Since the EMD forces all units from the dataset with fewer neurons to have an assigned match, we use vertical $z$-distance to threshold out the biologically impossible unit assignments. We then calculated the accuracy $R_{acc}$, i.e., the fraction of EMD unit assignments within the $z$-distance threshold which agree with the reference assignments.

$$P\left(\left(EMD \mid ref\right) \cap threshold\right) = \frac{P\left(\left(EMD \cap ref\right) \mid threshold\right)}{P\left(ref \mid threshold\right)} \tag{2}$$

We also retrieved non-reference units, i.e., matched units without receptive field information but whose $z$-distance is smaller than the threshold.

Second, we tracked units between consecutive datasets and summarized and analyzed the waveforms, unit locations, firing rates, and visual responses (see *Figure 5—figure supplement 1* to Figure 5 for details) of all tracked chains, i.e., units which can be tracked across at least three consecutive datasets.

## Measuring rigid drift using the EMD

Drift happens mostly along the direction of probe insertion (vertical or $z$-direction). We want to estimate the amount of vertical drift under the assumption that part of the drift is rigid; this is likely a good assumption given the small ($\approx 720$ μm) $z$-range of these recordings. The EMD allows us to extract the homogeneous (rigid) movement of matched units. For ideal datasets with a few units consistently detected across days, this problem is relatively simple (*Figure 2a*). In the real data analyzed here, we find that only $\approx 60\%$ of units are detected across pairs of days, so the rigid motion of the real pairs must be detected against a background of units with no true match. These units with no real match will have $z$-shifts far from the consensus $z$-shift of the paired units (*Figure 2c*).

In *Figure 2* the EMD match of units from the first dataset (*Figure 2b*, open circles) to the dataset recorded the next day (*Figure 2b*, closed circles) is indicated by the arrows between them. To demonstrate detection of significant drift, we added a 12 μm upward drift to the $z$-coordinate of the units from the second day. The first stage of the EMD is used to find matches using the combined distance metric as described in section Calculating the EMD metric. We used a kernel fit to the distribution of $z$-distances of all matched units to find the mode (mode = 15.65μm); this most probable distance is the estimate of the drift (*Figure 2c*). It is close to the actual imposed drift ($d_i = 12$μm).

As the EMD is an optimization algorithm with no biological constraints, it assigns matches to all units in the smaller dataset regardless of biophysical plausibility. As a result, some of the assigned matches may have unrealistically long distances. A distance threshold is therefore required to select correct pairs. For the illustration in *Figure 2*, the threshold is set to 15 μm, which is chosen to be

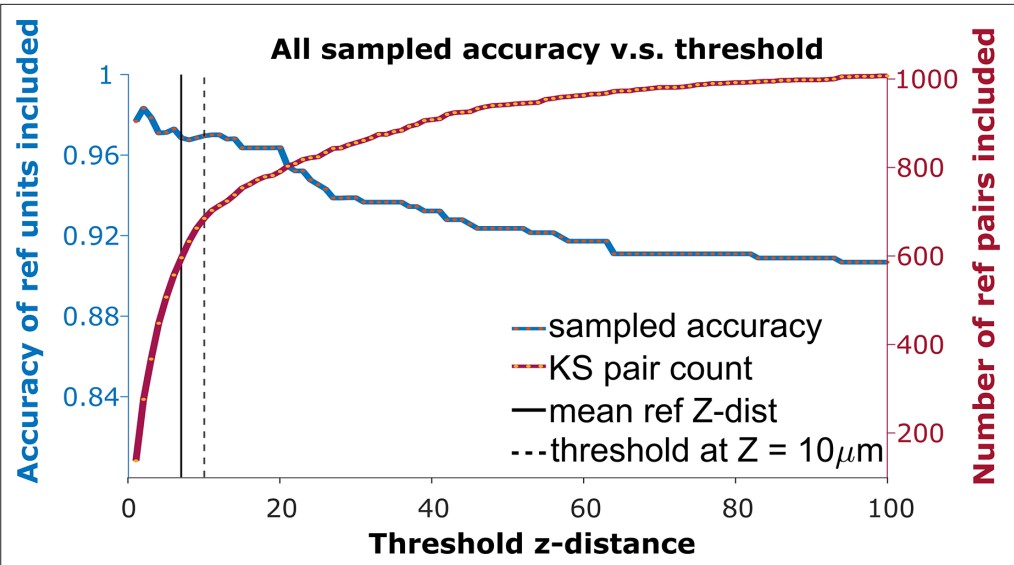

**Figure 3.** The receiver operator characteristic (ROC) curve of matching accuracy vs. distance. The blue curve shows the accuracy for reference units. The red line indicates the number of reference units included. The solid vertical line indicates the average z-distance across all reference pairs in all animals ($z = 6.96 \mu m$). The dashed vertical black line indicates a z-distance threshold at z=10 µm.

larger than most of the z-shifts observed in our experimental data. The threshold value will be refined later by distribution fitting (Figure 4). In *Figure 2* all of the sub-threshold (short) distances belong to upward pairs (*Figure 2b and c*, red solid arrows), showing that the EMD can detect the homogeneous movement direction and the amount of imposed drift.

When determining matched reference units from visual response data, we require that units be spatially nearby (within 30 µm) as well as having similar visual responses. After correcting for drift, we find that we recover more reference units (*Figure 2—figure supplement 1*), indicating improved spatial match of the two ensembles. This improved recovery provides further evidence of the success of the drift correction.

## A vertical distance threshold is necessary for accurate tracking

To detect the homogeneous z-shift of correct matches against the background of units without true matches, it is necessary to apply a threshold on the z-shift. When tracking units after shift correction, a vertical distance threshold is again required to determine which matches are reasonable in consideration of biological plausibility. The receiver operator characteristic (ROC) curve in *Figure 3* shows the fraction of reference units matched correctly and the number of reference pairs retained as a function of z-distance threshold. We want to determine the threshold that maximizes the overall accuracy in the reference units (*Figure 3*, blue curve) while including as many reference units as possible (*Figure 3*, red curve).

Since reference units only account for 29% of KSgood units (units with few ISI violations that are believed to represent a single unit), and the majority of KSgood units did not show a distinguishable visual response, we need to understand how representative the reference units are of all KSgood units.

We found the distribution of z-distances of reference pairs is different from the distribution of all KSgood units (*Figure 4a*, top and middle panels). While both distributions may be fit to an exponential decay, the best fit decay constant is significantly different (Kolmogorov-Smirnov test, reject $H0$, p $= 5.5 \times 10^{-31}$). Therefore, the accuracy predicted by the ROC of reference pairs in *Figure 3* will not apply to the set of all KSgood pairs. The difference in distribution is likely due to the reference units being a special subset of KSgood units in which units are guaranteed to be found in both datasets, whereas the remaining units may not have a real match in the second dataset. To estimate the ROC curve for the set of all KSgood units, we must estimate the z-distance distribution for a mixture of correct and incorrect pairs.

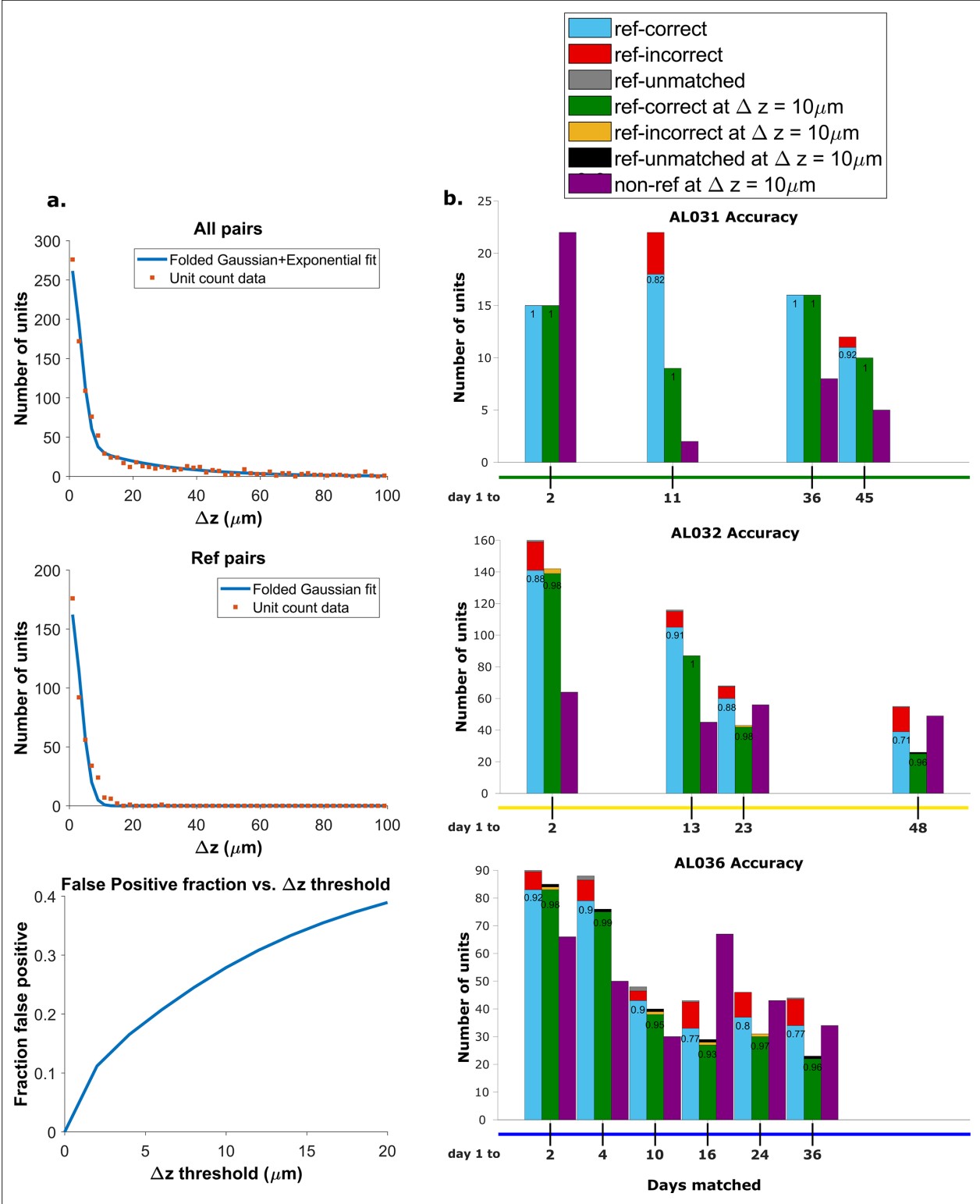

**Figure 4.** Recovery rate, accuracy, and putative pairs. (**a**) The histogram distribution fit for all KSgood units (top) and reference units alone (middle). False positives for reference units are defined as units matched by earth mover's distance (EMD) but not matched when using receptive fields. The false positive fraction for the set of all KSgood units is obtained by integration. $z$=10 μm threshold has a false positive rate = 27% for KSgood units. (**b**) Light blue bars represent the number of reference units successfully recovered using only unit location and waveform. The numbers on the bars are the recovery rate of each dataset, and the red portion indicates incorrect matches. Incorrect matches are cases where units with a known match from receptive field data are paired with a different unit by EMD; these errors are false positives. The green bars show matching accuracy for the set of

*Figure 4 continued on next page*

*Figure 4 continued*

pairs with *z*-distance less than the 10 μm threshold. The orange portion indicates incorrect matches after thresholding. The false positives are mostly eliminated by adding the threshold. Purple bars are the number of putative units (unit with no reference information) inferred with *z*-threshold=10 μm.

The online version of this article includes the following figure supplement(s) for figure 4:

**Figure supplement 1.** Determining the functional form for the *z*-distance distribution of all pairs.

**Figure supplement 2.** Fits of experimental *z*-distance distributions to the model.

**Figure supplement 3.** The reference unit recovery rate vs. days between matched recordings.

We assume that the distribution of *z*-distances $P(\Delta)$ for reference units is the conditional probability $P(\Delta \mid H)$; i.e., we assume all reference units are true hits. The distribution of *z*-distances for all KSgood units $P(\Delta)$ includes both hits and false positives. The distance distribution of false positives is the difference between the two.

A Monte Carlo simulation determined that the best model for fitting the *z*-distance distribution of reference units $P(\Delta \mid H)$ is a folded Gaussian distribution (*Figure 4a*, middle panel) and an exponential distribution for false positive units (see *Figure 4—figure supplement 1*). The KSgood distribution is a weighted combination of the folded Gaussian and an exponential:

$$P(AllUnits) = f * P(FoldedGaussian) + (1-f) * P(Exponential) \tag{3}$$

We fit the KSgood distribution to *Equation 3* to extract the individual distribution parameters and the fraction of true hits (*f*). The full distribution can then be integrated up to any given *z*-threshold value to calculate the false positive rate. (*Figure 4a*, bottom panel, see *Figure 4—figure supplement 2* for details).

Based on the the estimated false positive rate (*Figure 4a*, bottom panel), we used a threshold of 10 μm (*Figure 3*, black dotted line) to obtain at least 70% accuracy in the KSgood units. We used the same threshold to calculate the number of matched reference units and the corresponding reference unit accuracy (*Figure 4b*, green bars).

Note that this threshold eliminates most of the known false positive matches of reference pairs (*Figure 4b*, red fraction) at the cost of recovering fewer correct pairs (*Figure 4b*, green bars). The recovery rate varies from day to day; datasets separated by longer times tend to have higher tracking uncertainty (*Figure 4—figure supplement 3*).

In addition to the units with visual response data, we can track units which have no significant visual response (*Figure 4b*, purple bars). All comparisons are between subsequent datasets and the day 1 dataset.

## Units can be tracked in discontinuous recordings for 48 days

To assess long-term tracking capabilities, we tracked neurons across all datasets for each mouse. *Figure 5* shows a survival plot of the number of unit chains successfully tracked over all durations. All units in the plot can be tracked across at least three consecutive datasets, a chain as the term is used here. We categorized all trackable unit chains into three types: reference chains, mixed chains, and putative chains. Reference chains have receptive field information in all datasets. Putative chains have no reference information in any of the datasets. Mixed units have at least one dataset with no receptive field information. There are 133 reference chains, 135 mixed chains, and 84 putative chains across all the subjects. Among them, 46 reference, 51 mixed, and 9 putative units can be followed across all datasets. We refer to them as fully trackable units. One example trackable unit in each group is shown in *Figure 6*, *Figure 6—figure supplement 1*, and *Figure 6—figure supplement 2*.

We hypothesize that the three groups of units are not qualitatively different from each other, i.e., all units are equally trackable. In order to check for differences among the three groups, we analyzed the locations, firing rates, waveforms, and receptive fields of the fully trackable units in the three groups: reference, putative, and mixed.

The spatial-temporal waveform similarity is measured by the L2 distance between waveforms (section Calculating the EMD metric). A Kruskal-Wallis test is performed on the magnitude of L2 change between all pairs of matched waveforms among the three groups. There is no statistical difference in the waveform similarity in reference, putative, and mixed units (*H*=0.59, p=0.75) (*Figure 5—figure*

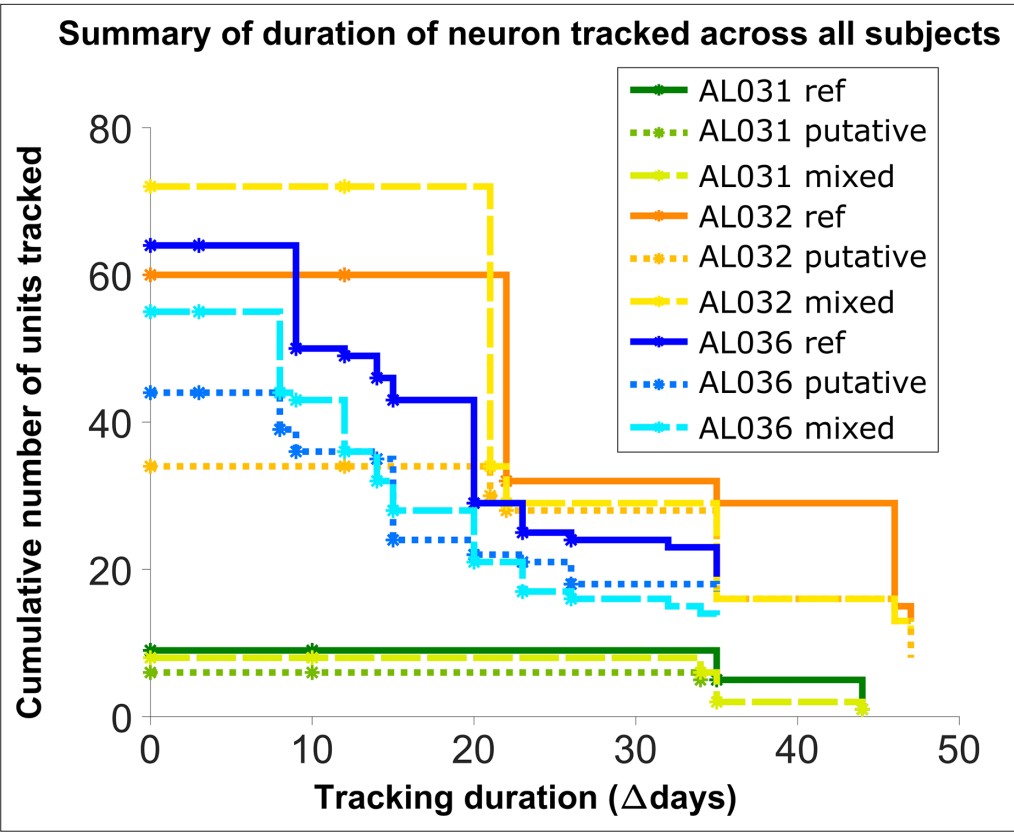

**Figure 5.** Number of reference units (deep blue, dark orange, and green for different subjects), putative (medium green, medium orange, and blue) units, and mixed units (light green, yellow, and light blue) tracked for different durations. The loss rate is similar for different chain types in the same subject. Note that chains can start on any day in the full set of recordings, so the different sets of neurons have chains with different spans between measurements.

The online version of this article includes the following figure supplement(s) for figure 5:

**Figure supplement 1.** Distribution of waveform L2 similarity change per dataset for each neuron group (reference, putative, and mixed) and across all neurons.

**Figure supplement 2.** Distributions of individual unit location changes over whole chains (top) and unit location changes between pairs of datasets (bottom), for each neuron group and across all neurons.

**Figure supplement 3.** Distribution of firing rate fold change per dataset for each neuron group and across all neurons.

**Figure supplement 4.** The visual fingerprint and peristimulus time histogram (PSTH) change distributions per dataset for each neuron group and across all neurons.

**Figure supplement 5.** The similarity score distribution per dataset for each neuron group and across all neurons.

---

*supplement 1*). There is no significant difference in the physical distances of units per dataset ($H$=1.31, p=0.52) (*Figure 5—figure supplement 2*, bottom panel), nor in the location change of units ($H$=0.23, p=0.89) (*Figure 5—figure supplement 2*, top panel).

Firing rate is characterized as the average firing rate fold change of each unit chain, with firing rate of each unit in each dataset normalized by the average firing rate of that dataset. There is no difference in the firing rate fold change in the three groups of units ($H$=1, p=0.6) (*Figure 5—figure supplement 3*).

The receptive field similarity between units in different datasets is described by visual fingerprint (vfp) correlation and peristimulus time histogram (PSTH) correlation between units, and the similarity score, the sum of the two correlations (section Reference set). The change in vfp between matched units is similar among the three groups ($H$=2.23, p=0.33). Similarly, the change in PSTH is not different among the three groups ($H$=1.61, p=0.45) (*Figure 5—figure supplement 4*).

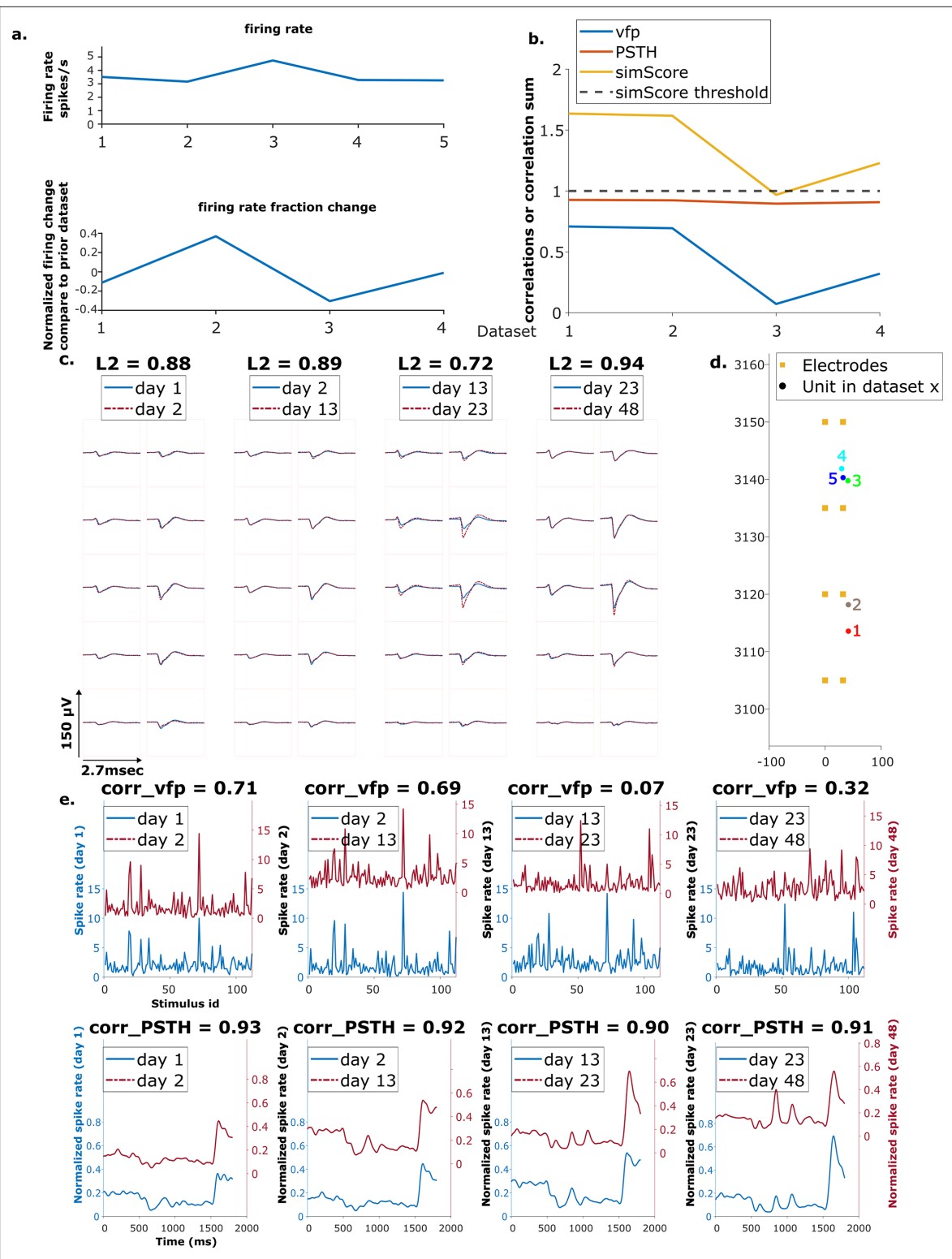

**Figure 6.** Example mixed chain. (**a**) Above: Firing rates of this neuron on each day (days 1, 2, 13, 23, 48). Below: Firing rate fractional change compared to the previous day. (**b**) Visual response similarity (yellow line), peristimulus time histogram (PSTH) correlation (orange line), and visual fingerprint (vfp) correlation (blue line). The similarity score is the sum of vfp and PSTH. The dashed black line shows the threshold to be considered a reference unit. (**c**) Spatial-temporal waveform of a trackable unit. Each pair of traces represents the waveform on a single channel. (**d**) Estimated location of this unit on different days. Each colored dot represents a unit on 1 day. The orange squares represent the electrodes. (**e**) The pairwise vfp and PSTH traces of this unit.

*Figure 6 continued on next page*

*Figure 6 continued*

The online version of this article includes the following figure supplement(s) for figure 6:

**Figure supplement 1.** Example reference chain.

**Figure supplement 2.** Example putative chain.

## Discussion

We present here an EMD-based neuron tracking algorithm that provides a new, automated way to track neurons over long-term experiments to enable the study of learning and adaptation with state-of-the-art high-density electrophysiology probes. We demonstrate our method by tracking neurons up to 48 days without using receptive field information. Our method achieves 90% recovery rate on average for neurons separated up to 1 week apart and 78% on average for neurons 5–7 weeks apart (*Figure 4b*, blue bars). We also achieved 99% accuracy up to 1 week apart and 95% 5–7 weeks apart, when applying a threshold of 10 μm (*Figure 4b*, green bars). It also retrieved a total of 552 tracked neurons with partial or no receptive field information, 12 per pair of datasets on average. All the fully trackable unit chains were evaluated by waveforms and estimated locations. Our method is simple and robust; it only requires spike sorting be performed once, independently, per dataset. In order to be more compatible and generalizable with existing sorting methods, we chose Kilosort, one of the most widely used spike sorting methods (*Sauerbrei et al., 2020*; *Stringer et al., 2019*). We show the capability of our method to track neurons with no specific tuning preference (*Figure 6—figure supplement 2*).

The method includes means to identify dataset pairs with very large drift. In our data, we can detect large drift because such datasets have very few reference units, and significantly different EMD cost. For example, datasets 1 and 2 in animal AL036 have very few reference units compared to other datasets (see *Figure 2—figure supplement 2*, AL036). This observation is consistent with the overall relationship between the EMD cost and recovery rate (*Figure 2—figure supplement 3*). Datasets with higher cost tend to have lower unit recovery rate and higher variation in recovery rates. Therefore, these two datasets were excluded in the tracking analysis.

Our validation relies on identifying reference units. The reference unit definition has limitations. The similarity score is largely driven by PSTHs (*Figure 7—figure supplement 1*), the timing of stimulus triggered response, rather than vfp, the response selectivity. As a result, a single neuron can be highly correlated, i.e., similarity score greater than 1, with more than 20 other neurons. For example, in subject AL032 shank 2, one neuron on day 1 has 22 highly correlated neurons on day 2, 4 of which are also within the distance of 30 μm. Non-reference units may also have very similar visual responses: we note that 33 (5 putative neurons and 28 mixed neurons) out of 106 trackable neurons have a similarity score greater than 1 even for days with no reference unit assignment. Coincidentally similar visual responses could potentially contribute to inaccurate assignment of reference units and irregularity in trackable unit analysis. These errors would reduce the measured accuracy of the EMD matching method; since the accuracy is very high (*Figure 4*), the impact of mismatches is low.

We note that the ratio of reference units over KSgood units decreases as recordings are further separated in time (*Figure 7*, *Figure 3*). This reduction in fraction of reference units might be partially due to representational drift as well as the fact that the set of active neurons are slightly different in each recording. The vfp similarity of matched neurons decreased to 60% after 40 days (see *Steinmetz et al., 2021* supplement).

We developed the new tracking algorithm based on an available visual cortex dataset, and used a prominent sorting algorithm (Kilosort 2.5) to spikesort the data. We had reference data to assess the success of the matching and tune parameters. Applying our algorithm in other brain areas and with other sorters may require parameter adjustment. Evaluation of the results in the absence of reference data requires a change to the fitting procedure.

The algorithm has only two parameters: the weighting factor $\omega$ that sets the relative weight of waveform distance vs. physical distance, and the $z$-distance threshold that selects matches that are likely correct. We found that recovery rate, and therefore accuracy, is insensitive to the value of $\omega$ for values larger than 1500 (*Figure 2—figure supplement 4*), so this parameter does not require precise tuning. However, the false positive rate is strongly dependent on the choice of $z$-distance threshold.

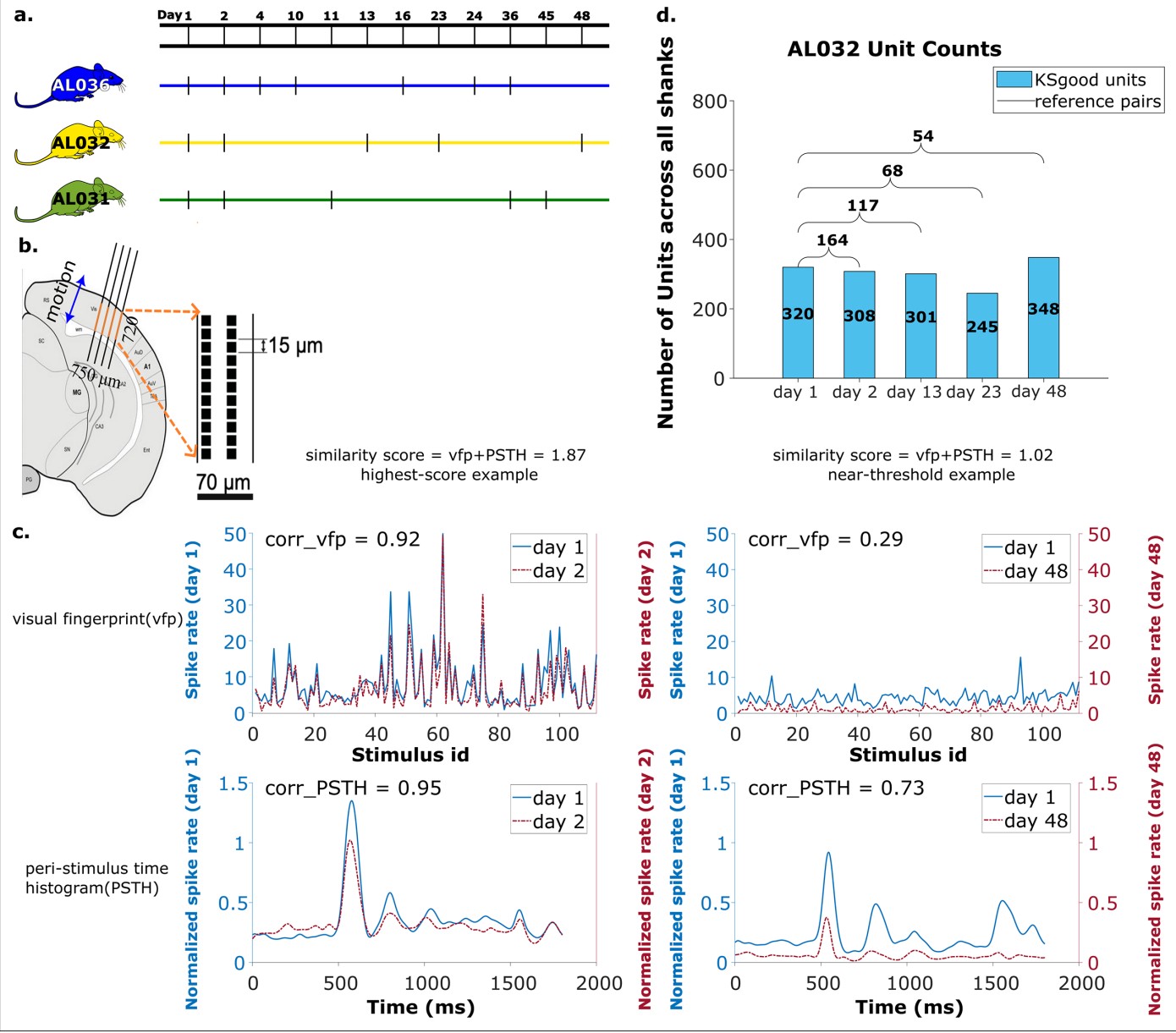

**Figure 7.** Summary of dataset. (**a**) The recording intervals for each animal. A black dash indicates one recording on that day. (**b**) All recordings are from visual cortex V1 with a 720 µm section of the probe containing 96 recording sites. The blue arrow indicates the main drift direction. (**c**) Examples of visual fingerprint (vfp) and peristimulus time histogram (PSTH) from a high correlation (left column) and a just-above-threshold (right column) correlation unit. Both vfp and PSTH values vary from [–1,1]. (**d**) Kilosort-good and reference unit counts for animal AL032, including units from all four shanks.

The online version of this article includes the following figure supplement(s) for figure 7:

**Figure supplement 1.** An example similarity score (visual fingerprint [vfp] + peristimulus time histogram [PSTH]) heatmap from animal AL032, shank 2, Kilosort-good units between days 1 and 2.

**Figure supplement 2.** The Kilosort-good and reference unit counts for the animals AL031 and AL036, as shown for animal AL032 in *Figure 7*.

**Figure supplement 3.** The ratio of the count of reference units to KSgood units decreases for pairs of datasets with larger time intervals.

When reference information (unit matches known from receptive fields or other data) is available, the procedure outlined in *Figure 4* can be followed. In that case, the distribution of *z*-distances of known pairs is fit to find the width of the distribution for correct matches. That parameter is then used in the fit of the *z*-distance distribution of all pairs to *Equation 3*. Integrating the distributions of correct and incorrect pairs yields the false positive rate vs. *z*-distance, allowing selection of a *z*-distance threshold for a target false positive rate.

In most cases, reference information is not available. However, the *z*-distance distributions for correct and incorrect pairs can still be estimated by fitting the distribution of all pairs. In *Figure 4— figure supplement 2* we show the results of fitting the *z*-distribution of all pairs without fixing the width of the distribution of correct matches. The result slightly underestimates this width, and the estimated false positive rate increases. This result is important because it suggests the accuracy estimate from this analysis will be conservative. We detail the procedure for fitting the *z*-distance distribution in Methods section (Algorithm 2).

As suggested in *Dhawale et al., 2017*, discontinuous recordings will have more false positives. Improving spike sorting and restricting the analysis to reliably sorted units will help decrease the false positive rate. Current spike sorting methods involve fitting many parameters. Due to the stochastic nature of template initialization, only around 60–70% units are found repeatedly in independently executed analysis passes. This leads to unpaired units which decrease EMD matching accuracy. Future users may consider limiting their analysis to the most reliably detected units for tracking; requiring consensus across analysis passes or sorters is a possible strategy. Finally, more frequent data acquisition during experiments will provide more intermediate stages for tracking and involve smaller drift between consecutive recordings.

## Methods

Our neuron tracking algorithm uses the EMD optimization algorithm. The minimized distance is a weighted combination of physical distance and 'waveform distance': the algorithm seeks to form pairs that are closest in space and have the most similar waveforms. We test the performance of the algorithm by comparing EMD matches to reference pairs determined from visual receptive fields (section Reference set). We calculate two performance metrics. The 'recovery rate' is the percentage of reference units that are correctly matched by the EMD procedure. The 'accuracy' is the percentage of correctly matched reference units that pass the *z*-distance threshold (*Figure 4a*). 'Putative units' are units matched by the procedure which do not have reference receptive field information. 'Chains' are units that can be tracked across at least three consecutive datasets. The full procedure is summarized in Algorithm 1.

**Algorithm 1** **Neuron matching procedure**

**Input:** channel map, unit cluster label, cluster mean waveforms (with $K_{loc} = 2$ and $K_{wf} = 5$
rows and $K_{col} = 2$ columns of channels), and spike times
**Step 1** Estimate unit locations
Estimate background amplitude for each unit
**for** all KSgood units $u_n \in U$ **do**
 **if** peak-to-peak voltage $V_{ptp} > 60\mu V$ **then**
 Get $u_n$'s waveform on channels $C_m$
 Get the peak-to-peak amplitudes $V_{ptp_c}$ of $u_n$ background-subtracted waveforms on
 channels $C_{u_n} = \{mc_{u_n} - k_{loc}, ..., mc_{u_n} + k_{loc}\}$, where $mc_{u_n}$ is the peak channel
 Estimate the neuron's 3D location as in **Boussard et al., 2021**;

$$f(x, y, z) = \sum_{c \in Cu_n} \left( V_{ptp_c} - \frac{1}{\sqrt{(x-x_c)^2 + (z-z_c)^2 + y^2}} \right)^2 \text{ where } x, z, \text{ and } y \text{ are the}$$

 horizontal locations, vertical location, and distance of the unit from the probe,
 respectively.
 Find an estimate of the global minimizer of $f, x_{u_n}, y_{u_n}, z_{u_n}$ using least-squares
 optimization
 **end**
**end**
**Step 2** Compute waveform similarity metrics
**for** waveforms $wf_{xi} \in U_{N1}$ and $wf_{yk} \in U_{N2}$, where $U_{N1}, U_{N2}$ are the set of all units in the
two datasets
**do**
 Centered at peak channel $mc_{xi}$ and $mc_{yk}$, respectively
 Get the sets of channels for each unit: $C_{u_n} = \{mc_{u_n} - k_{wf}, ..., mc_{u_n} + k_{wf}\}$
 There are $K_{wf} * 2 * K_{col} + 2 = 22$ channels for each unit
 Compute the waveform similarity metric as
 $(1/22) * \sum\limits_{c \in Cu_{xi}, Cu_{yk}} L2(wf_{xi} - wf_{yk}) / max(L2(wf_{xi}), L2(wf_{yk}))$
 for each of the 22 channels
**end**
**Step 3** Between-session drift correction
 Run the EMD with distances in physical and waveform space
 Estimate $z$-distance mode of all matched pairs with Gaussian kernel fit
 Apply correction on physical distances of all units $\in U_2 : z_{corr} = z - z_{mode}$
**Step 4** Unit matching
 Run the EMD with corrected physical distance and waveform metrics
 Set $z$-distance threshold to select unit pairs likely to be the same neuron
**Output:** cost $\sum d_{EMD}$, unit assignments

## Algorithm

### Earth mover's distance

The EMD is an optimization-based metric developed in the context of optimal transport and measuring distances between probability distributions. It frames the question as moving dirt, in our case, units from the first dataset, into holes, which here are the neural units in the second dataset. The distance between the 'dirt' and the 'holes' determines how the optimization program will prioritize a given match. Specifically, the EMD seeks to minimize the total work needed to move the dirt to the holes, i.e., neurons in day 1 to day 2, by solving for a minimum overall effort, the sum of distances (**Bertrand et al., 2020**; **Cohen, 1999**).

$$\min_{d_F} \sum_{ik} D(x_i, y_k), \text{ where } D = d_{loc} + \omega d_{wf}$$

$$\text{subject to } f_{ik} \in [0, 1] \; \forall i, k$$

$$\sum_k (f_k) \leq length(Y)$$

$$\sum_i (f_i) \leq length(X) \tag{4}$$

$$\sum (F) = min\left(\sum X, \sum Y\right)$$

in which $d_{loc} \in D^3$ is the 3D physical distance between a unit from the first dataset $x_i$, and a unit from the second dataset $y_k$. $d_{wf} \in D^1$ is a scalar representing the similarity between waveforms of units $x_i$ and $y_k$. $\omega$ is a weight parameter that was tuned to maximize the recovery rate of correctly matched

reference units. *F* is the vector of matched objects between the two datasets (see *Figure 2—figure supplement 4* for details about selecting weight).

The EMD has three benefits:

- It allows combining different types of information into the 'distance matrix' to characterize the features of units.
- The EMD can detect homogeneous movement of units (*Figure 2c*), thus providing a way for rigid drift correction, as described in section Between-session drift correction.
- By minimizing overall distances, the EMD has tolerance for imperfect drift correction, error in the determination of unit positions, and possible non-rigid motion of the units.

However, since the EMD is an optimization method with no assumptions about the biological properties of the data, it makes all possible matches. We therefore added a threshold on the permissible *z*-distance to select physically plausible matches.

## Calculating the EMD metric

The unit locations are estimated by fitting 10 peak-to-peak amplitudes from adjacent electrodes and the corresponding channel positions with a 1/*R* distance model (*Boussard et al., 2021*). Unlike *Boussard et al., 2021*, we operate on the mean waveforms for each unit rather than individual spikes. We found using the mean waveform yields comparable results and saves significant computation time. Unit locations are 3D coordinates estimated relative to the probe, where the location of the first electrode on the left column at the tip is considered the origin. The mean waveform is computed by averaging all the spike snippets assigned to the cluster by KS 2.5.

For 10 channels $c \in C_{u_n}$, find the location coordinates $x_{u_n}, y_{u_n}, z_{u_n}$ that minimizes the difference between measured amplitudes $V_{PTP}$ and amplitudes estimated with locations $\frac{\alpha}{\sqrt{(x-x_c)^2+(z-z_c)^2+y^2}}$:

$$min \sum_{c \in C_{u_n}} \left( V_{PTP_c} - \frac{1}{\sqrt{\left(x - x_c\right)^2 + \left(z - z_c\right)^2 + y^2}} \right)^2 \tag{5}$$

The locations are used to calculate the physical distance portion of the EMD.

For the waveform similarity metric, we want to describe the waveform characteristics of each unit with its spatial-temporal waveform at the channels capturing the largest signal. The waveform similarity metric between any two waveforms $u_{n1}$ and $u_{n2}$ in the two datasets is a scalar calculated as a normalized L2 metric (see Algorithm 1 step 2) on the peak channels, namely the channel row with the highest amplitude and 5 rows above and below (a total of 22 channels). The resulting scalar reflects the 'distance' between the two units in the waveform space and is used to provide information about the waveform similarity of the units. It is used for between-session drift correction and neuron matching. *Figure 1c* shows an example waveform of a reference unit.

## Between-session drift correction

Based on previous understanding of the drift in chronic implants, we assumed that the majority of drift occurs along the direction of the probe insertion, i.e., vertical *z*-direction. This rigid drift amount is estimated by the mode of the *z*-distance distribution of the EMD assigned units using a normal kernel density estimation implemented in MATLAB. We only included KSgood units (*Pachitariu et al., 2016*). The estimated drift is then applied back to correct both the reference units and the EMD matrix by adjusting the *z*-coordinates of the units. For validation, the post-drift correction reference set is compared with the post-drift correction matching results (from step 4 in Algorithm 1).

### Determining *z*-distance threshold

Determining the *z*-distance threshold to achieve a target false positive rate requires estimating the widths of the *z*-distance distributions of correct and incorrect pairs. If reference data is available, the *z*-distance distribution of the known correct pairs should be fit to a folded Gaussian as described in *Figure 4*. The width of the folded Gaussian, which is the error in determination of the *z*-positions of units, is then fixed in the fit of the *z*-distribution of all pairs found by the algorithm outlined in Algorithm 1. If no reference data is available, the width of the distribution of correct pairs is determined by fitting the *z*-distance distribution of all pairs to *Equation 3* with the folded Gaussian width as one

of the parameters. This procedure is detailed in Algorithm 2. We show two examples of model fitting without reference information in *Figure 4—figure supplement 2*.

---

Algorithm 2 **Determining an appropriate *z*-distance threshold**

---

**Input:** *z*-distances of all matched units, target false positive rate, width $\sigma$ of the *z*-distance distribution of correct pairs, if available

**Step 1** Fit *z*-distance distribution of all pairs to decompose into distributions of correct and incorrect pairs

Fit the *z*-distance distribution of all pairs to the sum of a folded Gaussian (for correct pairs) and an exponential (for incorrect pairs). If the width $\sigma$ of the distribution of correct pairs is known from reference data, fix at that value. Otherwise, include in the fit parameters. The functional form is: $P\left(z\right) = d\left(fNe^{-\frac{z^2}{2\sigma^2}} + \frac{1-f}{c}e^{-\frac{z}{c}}\right)$

where: $f$ = fraction of correct pairs; $\sigma$ = width of the distribution of correct pairs; $c$ = decay constant of distribution of incorrect pairs; $d$ = amplitude normalization; and $N = \frac{2}{\sigma\sqrt{2\pi}}$, the normalization factor of the folded Gaussian.

**Step 2** Determine *z*-threshold to achieve a target false positive rate

For Neuropixels 1.0 and 2.0 probes, the width of the *z*-distance distribution of correct matches ($\sigma$) should be <10 μm; a larger width, or a very small value of the fraction of correct pairs suggests few or no correct matches. In this case, the EMD cost is likely to be large as well (see *Figure 2—figure supplement 2* Animal AL036 first two rows). For a range of *z*-values, integrate the *z*-distance distribution of incorrect pairs from 0 to *z*, and divide by the integral of the distribution of all pairs over that range. This generates the false positive rate vs. *z*-distance threshold, as shown in *Figure 4—figure supplement 2*.

**Output:** $\sigma$ (uncertainty of position estimation), threshold at the target false positive rate

---

## Dataset

The data used in this work are recordings collected from two chronically implanted NP 2.0 four-shank probes and one chronically implanted one-shank NP 2.0 probe in the visual cortex of three head fixed mice (*Figure 7b*, see *Steinmetz et al., 2021*, for experiment details). The recordings were taken while 112 visual stimuli were shown from three surrounding screens (data from *Steinmetz et al., 2021*, Supplement Section 1.2). The same bank of stimuli was presented five times, with order shuffled. The four-shank probes had the 384 recording channels mapped to 96 sites on each shank.

We analyzed 65 recordings, each from one shank, collected in 17 sessions (5 sessions for animal AL031, 5 sessions for animal AL032, and 7 sessions for animal AL036). The time gap between recordings ranges from 1 to 47 days (*Figure 7a*), with recording duration ranging from 1917 to 2522 s. The sample rate is 30 kHz for all recordings. There are a total of 2958 KSgood units analyzed across all animals and shanks, with an average of 56 units per dataset (*Figure 7d* and *Figure 7—figure supplement 2*).

## Reference set

To track clusters across days, *Steinmetz et al., 2021*, concatenated two recording sessions and took advantage of the within-recording drift correction feature of Kilosort 2.0 to extract spikes from the 2 days with a common set of templates. They first estimated the between-session drift of each recording from the pattern of firing rate and amplitude on the probe and applied a position correction of an integer number of probe rows (15 μm for the probes used). Then two corrected recordings were concatenated and sorted as a single recording. This procedure ensured that the same templates are used to extract spikes across both recordings, so that putative matches are extracted with the same template. A unit from the first half of the recording is counted as the same neuron if its visual response is more similar to that from the same cluster in the second half of the recording than to the visual response of the physically nearest neighbor unit. Using this procedure and matching criteria, 93% of the matches were correct for recordings <16 days apart, and 85% were correct for recordings from 3 to 9 weeks (see *Steinmetz et al., 2021*, *Figure 4*). In addition, although mean fingerprint similarity decreases for recordings separated by more than 16 days, this decline is only 40% for the same unit recorded from 40 days apart (see *Steinmetz et al., 2021*, Supplement S3). This procedure, while successful in their setting, was limited to the use of integral row adjustments of the data for between-session drift correction and relied on a customized version of Kilosort 2.0. Although up to three recordings can be sorted together, they must come

from recording sessions close in time. In addition, a separate spike sorting session needs to be performed for every pair of recordings to be matched, which is time-consuming and introduces extra sorting uncertainty.

To find units with matched visual responses, we examine the visual response similarity across all possible pairs. The visual response similarity score follows *Steinmetz et al., 2021*, and consists of two measurements. (1) The PSTH, which is the histogram of the firing of a neuron across all presentations of all images, in a 1800 ms time window starting 400 ms before and ending 400 ms after the stimulus presentation. The PSTH is calculated by histogramming spike times relative to stimulus on time for all stimuli, using 1 ms bins. This histogram is then smoothed with a Gaussian filter. (2) The vfp is the average response of the neuron to each of the 112 images. The vfp is calculated by averaging the spike counts in response to each natural image from the stimulus onset to 1 s afterward across five shuffled trials.

Following *Steinmetz et al., 2021*, the similarity score between two neurons is the sum of the correlation of the PSTH and the correlation of the vfp across two sessions. The two correlations have values in the range (–1, 1), and the similarity score ranges from (–2, 2).

The pool of reference units is established with three criteria: (1) The visual response similarity score of the pair, as described above, is greater than 1 and their physical distance, both before and after drift correction, is smaller than 30 µm. A physical distance criterion is necessary, because some units have several potential partners with high visual response similarity (*Figure 7—figure supplement 1*). We impose the 30 µm threshold on both pre- and post-correction data because the drift is relatively small in our case, and we can reduce false positives by constraining the reference units to be in a smaller region without losing units. In general, one could apply the threshold only on corrected data (after drift correction). (2) A Kruskal-Wallis test is applied on all trials of the vfps to ensure the triggered response to the stimulus is significantly distinguishable from a flat line. (3) Select units from each recording that meet the good criteria in Kilosort. Kilosort assigns a label of either single-unit (good) or multi-unit to all sorted clusters based on ISI violations (*Pachitariu et al., 2016*). This step aims to ensure included units are well separated. If there are multiple potential partners for a unit, the pair with the highest similarity score is selected as the reference unit. The complete pool of reference units includes comparisons of all pairs of recordings for each shank in each animal. The portion of units with qualified visual response ranges from 5% to 61%, depending on the time gap between datasets (*Figure 7—figure supplement 3*). Overall, these reference units made up 29% of all KSgood units (*Figure 7—figure supplement 2*) across all three animals in our dataset. *Figure 7c* shows examples of visual responses from a high similarity reference unit and a reference unit with similarity just above threshold.

## Code sharing

All code used can be accessed at: https://github.com/janelia-TDHarrisLab/Yuan-Neuron_Tracking (copy archived at *Janelia-TDHarrisLab, 2024*).

## Acknowledgements

NIH grant U01 NS115587 in part supported TDH and AXY. We thank Claudia Böhm and Albert Lee for allowing us to use their data in Figure 4—figure supplement 2.

## Additional information

### Funding

| Funder | Grant reference number | Author |
| --- | --- | --- |
| BRAIN Initiative | U01 NS115587 | Augustine Xiaoran Yuan Timothy D Harris |

The funders had no role in study design, data collection and interpretation, or the decision to submit the work for publication.

## Author contributions
Augustine Xiaoran Yuan, Software, Formal analysis, Validation, Visualization, Methodology, Writing - original draft, Writing – review and editing; Jennifer Colonell, Conceptualization, Software, Formal analysis, Supervision, Validation, Visualization, Methodology, Writing - original draft, Writing – review and editing; Anna Lebedeva, Michael Okun, Data curation, Investigation; Adam S Charles, Conceptualization, Resources, Software, Formal analysis, Supervision, Funding acquisition, Visualization, Methodology, Project administration, Writing – review and editing; Timothy D Harris, Conceptualization, Resources, Supervision, Funding acquisition, Visualization, Project administration, Writing – review and editing

## Author ORCIDs
Jennifer Colonell ◉ http://orcid.org/0009-0009-3940-0689
Adam S Charles ◉ https://orcid.org/0000-0002-9045-3489
Timothy D Harris ◉ https://orcid.org/0000-0002-6289-4439

Reviewer #1 (Public Review): https://doi.org/10.7554/eLife.92495.3.sa1
Reviewer #2 (Public Review): https://doi.org/10.7554/eLife.92495.3.sa2
Author response https://doi.org/10.7554/eLife.92495.3.sa3

# Additional files

## Supplementary files
• MDAR checklist

## Data availability
Data may be found at: https://doi.org/10.5522/04/24411841.v1.

The following dataset was generated:

| Author(s) | Year | Dataset title | Dataset URL | Database and Identifier |
|---|---|---|---|---|
| Lebedeva A, Okun M, Krumin M, Carandini M | 2023 | Chronic recordings from Neuropixels 2.0 probes in mice | https://doi.org/10.5522/04/24411841.v1 | UCL, 10.5522/04/24411841.v1 |

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
