## [Editor Report · eLife assessment]

This **important** study proposes a new method for tracking neurons recorded with Neuropixel electrodes across days. The methods and the strength of the evidence are **convincing**, but the authors do not address whether their approach can be generalized to other brain areas, species, behaviors, or tools. Overall, this method will be potentially of interest to many neuroscientists who want to study long-term activity changes of individual neurons in the brain.

---

## [Referee Report · Reviewer #1 (Public Review)]

Neurons are not static-their activity patterns change as the result of learning, aging, and disease. Reliable tracking of activity from individual neurons across long time periods would enable studies of these important dynamics. For this reason, the authors' efforts to track electrophysiological activity across days without relying on matching neural receptive fields (which can change due to learning, aging, and disease) is very important.

By utilizing the tightly-spaced electrodes on Neuropixels probes, they are able to measure the physical distance and the waveform shape 'distance' between sorted units recorded on different days. To tune the matching algorithm and to validate the results, they used the visual receptive fields of neurons in the mouse visual cortex (which tend to change little over time) as ground truth. Their approach performs quite well, with a high proportion of neurons accurately matched across multiple weeks.

This suggests that the method may be useable in other cases where the receptive fields can't be used as ground truth to validate the tracking. This potential extendibility to tougher applications is where this approach holds the most promise. However, the study only looks at one brain area (visual cortex), in one species (mouse), using one type of spike sorter (Kilosort), and one type of behavioral prep (head-fixed). While the authors suggest methods to generalize their technique to other experimental conditions, no validation of those generalizations was done using data from different experimental conditions. Anyone using this method under different conditions would therefore need to perform such validation themselves.

---

## [Referee Report · Reviewer #2 (Public Review)]

The manuscript presents a method for tracking neurons recorded with neuropixels across days, based on the matching of cells' spatial layouts and spike waveforms at the population level. The method is tested on neuropixel recordings of the visual cortex carried over 47 days, with the similarity in visual receptive fields used to verify the matches in cell identity.

This is an important tool as electrophysiological recordings have been notoriously limited in terms of tracking individual neuron's fate over time, unlike imaging approaches. The method is generally sound and properly tested but I think some clarifications would be helpful regarding the implementation of the method and some of the results.

(1) Page 6: I am not sure I understand the point of the imposed drift and how the value of 12µm is chosen.

Is it that various values of imposed drift are tried, the EMDs computed to produce histograms as in Fig2c, values of rigid drifts estimated based on the histogram modes, and then the value associated with minimum cost selected? The corresponding manuscript section would need some clarification regarding this aspect.

(2) The EMD is based on the linear sum, with identical weight, of cell distance and waveform similarity measures. How performance is affected from using a different weighting of the 2 measures (for instance, using only cell distance and no waveform similarity)? It is common that spike waveforms associated to a given neuron appear different on different channels of silicon probes (i.e. the spike waveform changes depending the position of recording sites relative to the neuron), so I wonder if that feature is helping or potentially impeding the tracking.

(3) Fig.5: I assume the dots are representing time gaps for which cell tracking is estimated. The 3 different groups of colors correspond to the 3 mice used. For a given mouse, I would expect to always see 3 dots (for ref, putative and mixed) for a given tracking gap. However, for mouse AL036 for instance, at tracking duration of 8 days, a dot is visible for mixed but not for ref and putative. How come this is happening?

(4) Matched visual responses are measured by the sum of correlation of visual fingerprints, which are vectors of cells' average firing rate across visual stimuli, and correlation of PSTHs, which are implemented over all visual stimuli combined. I believe that some information is lost from combining all stimuli in the implementation of PSTHs (assuming that PSTHs show specificity to individual visual stimuli). The authors might consider, as alternative measure of matched visual responses, a correlation of the vector concatenations of all stimulus PSTHs. Such simpler measure would contain both visual fingerprint and PSTH information, and would not lose the information of PSTH specificity across visual stimuli.

2nd revision

(1) From reading the authors' response, I could understand several of the points I had previously missed. I still think that some part of the results are not straightforward to understand, the way it is written. Adding a few introductory sentences to the paragraphs (for instance the one related to my previous point #1) would really help the reader comprehend this important work.

(2) Following on my point #2, the w value used is 1500 and the recovery rate doesn't seems to reach a peak but rather a plateau for larger w values. From such large w value and the absence of a downward trend for increasing values, it would seem that only the 'waveform distance' matter and that the 'location distance' doesn't contribute much to the EMD distance. Is this correct?

---

## [Author Response]

The following is the authors’ response to the original reviews.

Reviewer #1, in both the public review and recommendations to authors, raises the important question of generalizability of the new technique to other brain areas, to analysis with sorters other than Kilosort, and in the absence of reference data. Specifically, how can experimenters working in brain areas other than visual cortex understand if the tracking is functioning, and set the parameters in the tracking pipeline.

We agree that generalizability of the tracking procedure is a serious issue, especially with respect to other brain areas with varying degrees of measured waveform preservation over time. As the number of potential recording conditions is combinatorial to experimentally test, we instead address these issues in the manuscript by providing a general prescription for interpreting the distribution of vertical distances of matched pairs that can be used for data from any recording using any spike-sorter (Methods section 4.2, Supplement section 8.4, figure S9, paragraphs 7-10 of the Discussion section). This extension of the method allows users to estimate the matching success in the context of their own data, even in the absence of reference data. To address the concern of overfitting, we have also added discussion covering adjustment of the two parameters in the procedure (the relative weight of waveform distance vs. physical distance, and the threshold for accepting matches as real) to the Discussion section.

Reviewer #2 suggested clarification of the following points in the public review. We answer those here and have also clarified these points in the main text where appropriate.

(1) What is the purpose of testing the drift correction with imposed drift (Figure 2, page 6 in the original manuscript), and how the value was chosen?

To test the ability of EMD to detect substantial drift, we need examples that resemble experimental data, including error in fit unit positions and units with no correct matches. We chose to create these examples by taking waveform and position sets from real data with modest drift, and adding a fixed shift to one dataset. The value of 12 um in the figure is arbitrary, simply an example in the range of real drift. These tests allow us to demonstrate the success of EMD for detection of drift in real data.

(2) How is performance affected by using a different weighting of the 2 measures (physical distance and waveform distance) in the EMD?

Recovery rate (number of reference units successfully matched in EMD) vs weighting of the waveform distance is shown in Supplement section 8.10. Recovery rate increases with low values of waveform weighting, leveling off at a value of 1500. We selected that inflection point for the analysis in this paper, to avoid coincidental matching of physically distant units with similar waveforms.

(3) Should the intervals measured in the survival plot in Figure 5 be identical for the three different classes of tracked neurons?

The plot includes all chains of tracked neurons, which can start on arbitrary days in the set of all recordings (see the definition of chains in section 2.4). As a result, the gaps between days, which determine where there is a point on the plot, can be different for the three sets of neurons (reference, putative, and mixed). We have added a comment to the Figure 5 caption to ensure this is clear.

(4) Would other metrics of the similarity of visual responses work better?

The similarity metric we use was adopted from the original paper using this data (reference 7). We chose to use the same metric both to take advantage of the original authors’ expertise about the data and allow for reasonable comparison of the new technique to theirs. It is correct that this similarity metric alone does not allow for unique matching (see Discussion and Supplement section 8.2). However, the agreement of EMD with reference pairs determined from the combination of position and visual response similarity is very high, suggesting there are few incorrect reference pairs. Any incorrect reference pairs cause an underestimate of the tracking accuracy.

(5) Add a definition of ROC.

Added this definition to the text.

**Reviewer #1 Recommendation to authors:**
The main text needs proofreading.

We agree that the manuscript needed more thorough proofreading, and we have made corrections of typos and minor language errors throughout.

Additional comment from the authors:

Since the posting of this manuscript, another method for tracking neurons has been introduced:

Enny H. van Beest, Célian Bimbard, Julie M. J. Fabre, Flóra Takács, Philip Coen, Anna Lebedeva, Kenneth Harris, Matteo Carandini, Tracking neurons across days with high-density probes, bioRxiv 2023.10.12.562040; doi: https://doi.org/10.1101/2023.10.12.562040